# Polyethylene Wax Modified by Organoclay Bentonite Used in the Lost-Wax Casting Process: Processing−Structure−Property Relationships

**DOI:** 10.3390/ma13102255

**Published:** 2020-05-14

**Authors:** Dorota Czarnecka-Komorowska, Krzysztof Grześkowiak, Paweł Popielarski, Mateusz Barczewski, Katarzyna Gawdzińska, Mikołaj Popławski

**Affiliations:** 1Polymer Processing Division, Institute of Materials Technology, Poznan University of Technology, 3 Piotrowo St., 61-138 Poznan, Poland; Mateusz.Barczewski@put.poznan.pl; 2Division of Foundry, Institute of Materials Technology, Poznan University of Technology, 3 Piotrowo St., 61-138 Poznan, Poland; Krzysztof.Grzeskowiak@put.poznan.pl (K.G.); Pawel.Popielarski@put.poznan.pl (P.P.); 3Department of Machines Construction and Materials, Maritime University of Szczecin, 2–4 Willowa St., 71-650 Szczecin, Poland; k.gawdzinska@am.szczecin.pl; 4Faculty of Engineering Materials and Technical Physics, Poznan University of Technology, 24 Jan Pawel II St., 61-139 Poznan, Poland; Mikolaj.Poplawski@put.poznan.pl

**Keywords:** modification, waxes, bentonite, wax/bentonite blends, viscosity, shrinkage, DSC, hardness, SEM, FT-IR spectroscopy, lost wax casting, alloy cast

## Abstract

In this paper, the influence of bentonite (BNT) on rheological properties, linear shrinkage, hardness, and structure of polyethylene wax (PE-wax) used in the lost-wax casting process are studied. Experiments were conducted with PE-wax with different contents of bentonite (from 0.1 to 5 wt.%). The structural changes of modified polyethylene wax were evaluated using differential scanning calorimetry (DSC) and scanning electron microscopy (SEM). The addition of bentonite in a concentration range between 0 and 0.8 wt.% into PE-wax caused a decrease of the linear shrinkage and the hardness of PE-wax/BNT blends. Modified polyethylene wax containing greater than 1 wt.% of the filler showed an increase in viscosity. The DSC-based investigations confirmed good miscibility and a uniform structure of PE-wax with BNT. Moreover, an increase in melting temperature onset and its enthalpy observed for filler PE-wax material showed an influence of bentonite on the crystalline structure of the polymeric matrix. SEM observations of the fractured surfaces confirmed the homogeneous structure of blends with contents up to 0.8 wt.% BNT. In the case of higher filler concentrations, the presence of more numerous and large clusters of filler in the polymer matrix was observed due to the increase in the viscosity of the PE-wax/BNT melt composition during mixing. The PE-wax blend with 0.4 wt.% BNT gives better results of reduced linear shrinkage and lower hardness than unmodified material. Lastly, a new developed material (PE-wax/0.4 BNT) was subjected to technological tests, consisting of the preparation of the mold and manufacture of a high-quality aluminum cast, using the lost-wax method.

## 1. Introduction

The production of castings by the method of lost-wax models consists of creating around the model, usually a ceramic mold, and then removing wax by smelting. The next stage is the process of pouring the finished mold with liquid metal and cleaning the resulting casting [1,2,3,4]. In the industry, for the described technology, there is a large variety of model masses, which are mostly waxes [5,6,7,8,9,10].

Wax, as the material used for the production of castings, has been known for millennia, and the type of wax used has depended primarily on the availability of the raw material and material cast [11,12]. There are many known divisions of waxes, as systematized in Figure 1. In addition, Wolff [11] divided waxes into natural waxes and waxes derived from petroleum waxes, and Marszałek et al. [12] divided waxes into natural and synthetic waxes.

Chakravorty et al. [13] proposed the division of waxes into three groups: natural waxes (waxes from bees, candelilla, carnauba, rice, cane), hydrocarbon waxes (ozocerite, ceresin, paraffine), and synthetic waxes (resins, amide, castor wax), as well as unfilled waxes and filled waxes, which, despite being non-Newtonian liquids [14], usually show similar behavior.

Currently, waxes are used in the foundry industry due to the increasing demands on finished products and the technological process. The creation of wax blends is designed to eliminate or maximally reduce the occurrence of detrimental phenomena, such as uncontrolled contraction or low strength. Wax mixtures, when manufacturing detailed precision castings, are virtually single use. After melting the wax from the ceramic mold, it can only be used to make less-detailed parts of the model, e.g., for the construction of fillers. However, it cannot be used again for the production of the model mass because there is a danger that it contains particles of foreign materials or ceramic mass. This may lead to unevenness on the surface of the model or to other surface defects, which can cause shortcomings [4,10,15,16,17,18,19].

Appropriate selection of individual components of the mixture is directly related to the course of the model formation process—the behavior of wax both in liquid and solid state. Depending on the expected effect, such blends usually consist of waxes, resins, plastics, and additives such as fillers or plasticizers [16]. Furthermore, there is a different division of the components of wax blends into straight or unfilled molding compounds, filled waxes, emulsified fillers, adhesive waxes for the assembly of wax patterns, and patching and repair waxes, as well as mercury pattern compounds and ice pattern compounds [17]. As it follows from the above, modern waxes are complex blends of many components that have a direct impact on the formation of the foundry model. Therefore, the development and application of the right mixture and the appropriate selection of individual parameters of the wax model formation process should form the basis for the correct execution of castings using the smelted model method.

There have been many publications in recent years on the subject of properties and waxes processing used in the foundry models [18,19,20,21,22,23]. One researcher who noted a lack of repeatability of the properties of waxes used in the foundry models was Okhuysen et al. [18]. By using computer prototyping to predict the dimensions of wax models, he found that wax shrinkage represented the largest share in the overall change in dimensions (comparing the finished cast to the wax model) [18]. Gebelin and Joly [19] showed that the accuracy of the wax model has a direct impact on the accuracy of the final casting. Bonilla et al. [20] and Singh et al. [21] independently researched the impact of the technological process (wax injection molding) on the dimensional and surface precision of the wax model. It has been proven that the dimensions of the final wax model are significantly influenced by the process parameters [20,21]. Others who researched the dimensional stability of waxes based on the parameters of the injection process were Rezavand et al. [22]. They chose the injection temperature and holding time as various processing parameters and concluded that the final dimensions of the wax pattern are affected by the type of wax, the geometry of the part, and process parameters. In contrast, Horacek et al. [23] showed the impact of the mutual relationship of injection parameters, such as injection temperature, die temperature, injection force, and holding time, on the final dimensions of the wax model.

The wax models (created from wax blends) are characterized with several essential, from a technological point of view, properties, such as dimensional accuracy, hardness, elasticity, wettability of the first coat, and surface smoothness [11,12,13,24]. Furthermore, Chakravorty at al. [13] also pointed to some disadvantages of wax models, i.e., core break, bowing, wall displacement, flow lines (hot and cold), non-fill (misrun), air entrapment, sinkage (cavitation), poor surface finish, sticking, distortion (due to residual stress and its relief) [25,26]. By properly selecting the components of the wax mixture, the occurrence of individual wax model defects may be reduced. A crucial aspect is to create a mixture whose individual components, while minimizing the occurrence of the selected defect, do not intensify the appearance of another. In addition, such blends should have a price that is acceptable for large-scale production realized in industrial conditions.

Fillers are playing an increasingly important role with the main task to modify the properties of the original blends. The addition of a filler may improve tensile or compressive strength, thermal and dimensional stability, hardness, density, surface quality, chemical resistance, and dielectric strength, or reduce the shrinkage of the material of the model, its abrasion, resistance to thermal shock, absorption, and degree of wear [13,15,17,27].

Organic fillers are among the most popular because, after burning, they leave a minimal amount of easy-to-remove ash. They have a low molecular structure, thanks to which they do not adversely affect the surface quality, and due to a similar specific weight since waxes ensure minimal phase separation during heating [9]. Examples of both inorganic and organic fillers used to create blends are polystyrene, organic acids, urea, bisphenol A-based materials, isophthalic acids, cross-linked polystyrene, silica, natural resins, soybean flour, wood flour, glass, clay, calcium carbonate, talc, graphene oxide, limestone, synthetic polymers, and bentonite [11,15,16,24,27,28].

Bentonite belongs to the group of inorganic fillers. It is a sedimentary clay rock, consisting mainly of montmorillonite (at least 75%) and a large number of minerals. It is used as a cleansing and decolorizing agent, support material (e.g., for the production of molding sand, drilling mud, and sewer drainage systems), and is also used in pharmacology. There are two types of bentonite: bentonite Wyoming (Na-bentonite) and meta bentonite (Ca-bentonite). The main properties of bentonite include high absorption, gelation, water dispersion, and ion exchange, especially by calcium (Ca) and magnesium (Mg) ions [15,27,29].

Bentonite is also often applied as a filler for both thermoplastic, thermoset polymer composites, and elastomers [30,31,32,33,34,35,36]. Since bentonite is hydrophilic, it is often not compatible with most polymers and can be chemically modified to render its surface to be more hydrophobic [33,35]. Additionally, chemical modification of bentonite becomes necessary to increase the polarity of the polymer matrix by grafting polar groups [36]. The improvement of the interfacial bonding between the hydrophilic fillers and the hydrophobic polypropylene matrix has also been an essential issue in the research field because the interfacial adhesion between the filler and polypropylene plays an important role in determining the properties of composites.

The process of dispersing bentonite in the polymer matrix is strongly dependent on the polymer clay compatibility because the processing of polymer blends necessitates knowledge of their rheological properties (viscosity). Because the rheological properties of particulate suspensions are responsive to the feature of the dispersed phase, they provide information about the internal microstructure of nanocomposites, such as the degree of dispersion of clay and the confinement effect of silicate layers on the motion of polymer chains [23,26,32,37].

There are numerous papers in which bentonite played the role of a filler or a modifier of the properties and structure of the polymers [15,16,27,31]. Widjijono et al. [15] studied melting behavior, hardness, and linear thermal expansion of wax blends with different concentrations of Ca-bentonite used as a filler. They showed that the mixture containing 20 wt.% filler had a 2 °C higher melting point and higher hardness with lower linear thermal expansion (0.17%, compared to 0.44% for unmodified wax) [15]. In another paper [27], Widjijono et al. showed that, for wax blends consisting of paraffin wax, Carnauba wax, and beeswax with different concentrations of Ca-bentonite filler, an increase in the amount of filler causes an improvement in the hardness of the wax blend in an almost linear manner. Bemblage et al. [16] studied the effect of proper selection of waxes without the use of fillers on the properties of mixtures. By creating their mixture containing 50% paraffin wax, 30% beeswax, and 20% Montana wax, they obtained the most beneficial results regarding the linear and general shrinkage of these blends and the best surface quality of finished products. In addition, they indicated the optimal parameters of the wax injection process as 68 °C (injection temperature), 48 °C (die temperature), 490 N (injection force), and 10 min (holding time) [16].

In the case of wax models, which arise as a result of solidification of the liquid mixture, the crystallization process has a significant impact on the final properties and structure, the course of which varies significantly due to the chemical composition of the mixture and the cooling time [38,39]. As a result, in dependence on the cooling rate, wax crystals are formed primarily in the form of lamellas, needle structures, or small-crystalline domains [40,41]. One of the critical parameters of the wax crystallization process is the wax appearance temperature (WAT), i.e., the temperature limit at which crystal formation begins. There are many methods for determining the WAT temperature, including density and viscosity, differential scanning calorimetry, chromatography, optical microscopy and X-ray diffraction [31,32,42,43].

The current research aimed to develop a new polyethylene-based wax mixture with low linear shrinkage and better thermal properties to produce a wax pattern, and then to make a good quality aluminum alloy cast. The effect of bentonite on the linear shrinkage, thermal stability, viscosity, morphology, chemical structure, and hardness properties of polyethylene (PE) wax used in the precision casting process was investigated. The aim of this paper was also to develop the possibilities of applying the new material, i.e., polyethylene wax modified by bentonite, to produce models in the foundry.

## 2. Experimental

### 2.1. Materials

The wax used was commercial injection wax (Flexible Blue) manufactured by Freeman Manufacturing & Supply (Avon, OH, USA), with a density of 0.945 g/cm^3^. Wax usually used as a matrix for manufacturing models in the lost-wax casting method is characterized by long shelf life, high flexibility, and ability for reproducing very sharp details. It is based on a mixture of wax and polyethylene and is especially well-suited for metal moulds and stone-in-place casting. Table 1 shows the selected physical properties of the wax according to the technical datasheet [44]. The PE-wax was modified with different bentonite powder content (abbreviated BNT).

The used particle-shaped filler in this study was commercially available bentonite, grade bentonite “SN” (Górniczo-Metalowe S.A. “Zębiec”, Starchowice, Poland) with a density of 0.816 g/cm^3^. Figure 2a illustrates Scanning Electron Microscope (SEM) pictures of the initial surface morphology of the bentonite made using secondary electron (SE) signals and an image prepared with a back-scattered electron (BSE) detector (TESCAN, Brno, Czech Republic) (Figure 2b). The chemical analysis of this filler is shown in Table 2. Furthermore, the elemental composition analysis (energy dispersive spectroscopy, EDS) mapping (a) along with the distribution of the chemical composition (b) of BNT are shown in Figure 3. Moreover, Figure 3 shows the SEM image and the distributions of silicon (Si), aluminum (Al), iron (Fe), and oxygen (O) elements of the section of bentonite by EDS. The BNT used in this study was characterized by a high content of Si (about 30 wt.%) and Al (10 wt.%). It also contained calcium (Ca) (2 wt.%) and magnesium (Mg) (2 wt.%) as well as trace amounts of sodium (Na).

Bentonite powder during preliminary investigations was characterized with a laser particle sizer (Fritsch ANALYSETTE 22 apparatus, Idar-Oberstein, Germany) operated in the range of 0.08–2000 µm. Figure 4 presents the cumulative size distribution Q3(x) and histogram dQ3(x) as a function of particle size. The results indicate that the used powder has no particles with a size above 100 μm. Moreover, the 50% passing size of the sample is 21 μm, and more than 90% of this material is below 48 μm.

### 2.2. Preparation of the Wax/Bentonite Blends

Bentonite was dried in an air oven at 100 °C for 24 h to expel moisture prior to blending with wax. Test models were made of PE-wax filled with various content of the BNT. The bentonite in powder form as filler under different concentrations, between 0 and 5 wt.%, was added to the PE-wax matrix, as shown in Table 3.

Three samples were prepared for each batch of wax/BNT. Values from the three measurements were averaged. The wax blends were held at 90 °C for 4 h. The melting was carried out in a resistance furnace type N 150/WAX manufactured by Nabertherm GmbH (Lilienthal, Germany). The ingredients were mixed mechanically at 90 °C for 10 min using an electric laboratory mixer with a rotational speed of 200 rpm. The models were made in demountable forms of silicone rubber (Gumosil AD, Silikony Polskie, Nowa Sarzyna, Poland) using an Olimpia WW-25 wax injection molding machine (Olimpia, Poland). The injection process parameters used during the manufacturing of the wax are shown in Table 4. Figure 5 shows the wax pattern and wax patterns assembled on a “tree.” The wax pattern is shown in Figure 5a, while wax/0.4BNT models were combined with the main inlet into a model set (Figure 5b). The elements were joined as a result of melting the elements of the filler system (main filler and supply inlets).

The experimental molds were built using the lost-wax casting technique in ceramic gypsum compounds. Casting molds were made of gypsum mass (elastic jewelry gypsum) poured into a steel sleeve with a diameter of 80 mm and a length of 120 mm. The gypsum mass was degassed in a degassing chamber at a pressure of 20 kPa for 5 min. After solidification (setting and hardening) of gypsum, the process of wax smelting and heat treatment (annealing) of molds was carried out. During heat treatment, chemically not bound, and bound water needs to be removed from casting molds together with residues of the compound. The course of the temperature program applied during the thermal treatment of the molds is presented in Figure 6.

### 2.3. Preparation of the Aluminum Cast

The castings were made in the centrifugal casting process. The prepared molds were removed from the furnace and poured with liquid aluminum alloy (AlSi7) at 750 °C. The centrifugal casting process uses the effect of centrifugal force on the flow of liquid metal in the mold with a resulting increase of the liquid and solidifying the metal pressure. Centrifugal force is more significant than gravity, and acting on liquid metal allows the correct metallostatic pressure to fill the mold with molten metal appropriately. The gypsum compound was carefully rinsed once the metal was solid and inspected to identify the mold portions filled with metal.

## 3. Characterization Methods

### 3.1. Viscosity and Density of Wax/Bentonite Blends

The viscosities of the wax filled with various amounts of bentonites were determined using a rotational rheometer (Anton Paar MCR 301, Anton Paar GmbH, Graz, Austria) at 71.1 °C (160 °F), which corresponds to the lowest temperature value used by the producer to determine the viscosity of the material presented in the technical specification. The measurements of the viscosity were realized in a constant shear mode using a 10 s^–1^ shear rate. After preliminary pre-shearing was realized for 60 s with a shear rate of 10 s^–1^ and 60 s relaxation before measurement, the constant shear experiment was realized using a 25-mm parallel plate measuring system with a gap of 0.3 mm. Additionally, the viscosity curves were evaluated in a cone-plate of a 25-mm measuring set up in the shear rate range of 0.01 to 100 s^–1^. The solid masses of the wax pure and wax/bentonite blends were measured by an electronic balance (AXIS AD50-AD200, AXIS, Gdansk, Poland). The density was measured based on PN-EN ISO 1183-1:2005 standards [45]. Ethyl alcohol (as an immersion liquid) was used, and measurements were made for five samples from each series.

### 3.2. Linear Shrinkage of Wax/Bentonite Blends

The waxes were gravitationally cast at 90 °C into metal molds with a diameter of 100 mm and height of 22 mm. After filling the molds, each sample and mold diameter was measured. The time between filling the molds and measurements of sample diameters was about 6 h. The linear shrinkage (*LS*) was measured based on the PN-EN ISO 2577 standard [46]. *LS* was calculated by measuring the difference between mold dimensions and pattern dimensions produced using Equation (1).
(1)$$LS=\frac{D_{M}-D_{S}}{D_{M}}\times 100\%$$
where *D_M_*—inner diameter of a mold, mm, *D_S_*—outer diameter of a sample, mm [46].

### 3.3. Thermal Properties (DSC) of Wax/Bentonite Blends

Differential scanning calorimetry (DSC) is a thermal analysis technique showing how physical properties of a sample change along with temperature against time. Melting and crystallization behavior of the wax/BNT blends were studied with a differential scanning calorimeter (Netzsch, DSC 204 F1 Phoenix, NETZSCH GmbH, Selb, Germany), operating under nitrogen flow (150 mL/min). Samples of about 10 mg were first heated from room temperature to 90 °C at 20 °C/min and held there in order to eliminate the thermal and mechanical prehistory. Next, the samples were cooled to 20 °C at a cooling rate of 10 °C/min and held there for 5 min and then reheated again to 90 °C at the same rate. Values of crystallization and melting entropies (ΔH_c_ and ΔH_m_) were calculated from the exothermic and endothermic peak areas and exothermic and endothermic peak temperatures were taken as crystallization and melting temperatures (T_c_ and T_m_), respectively. The second heating for each sample was used to evaluate the enthalpy of melting. The degree of crystallinity (weight fraction crystallinity) of partially crystalline polymers is influenced by the chemical structure and thermal history such as the cooling conditions during processing or post-thermal treatment [47].

Degree of crystallinity (*X_c_*) of the PE-wax samples was determined from the values of melting enthalpy, using Equation (2).
(2)$$X_{c}=\frac{\Delta H_{m}}{\left(1-\phi \right)\Delta H_{m}^{0}}\times 100\%$$
where Δ*H_m_* is the melting enthalpy of the samples (J/g), *ϕ* is the weight fraction of bentonite, and Δ*H*^0^*_m_* is the melting enthalpy for a 100% crystalline of the polymer, where this value was taken as 293 J/g for polyethylene [47].

### 3.4. Mechanical Properties (Shore D Hardness)

The mechanical properties of the wax modified with bentonite were determined by the Shore D hardness test. The hardness was measured based on the PN-EN ISO 868 standard [48] using a Shore durometer (Sauter HBD 100-0 GmbH, Balingen, Germany).

### 3.5. Scanning Electron Microscope (SEM)

The morphology of the fractured surfaces of PE-wax/BNT samples coated with a thin layer (20 nm) of carbon was analyzed using a scanning electron microscope (MIRA 3, TESCAN, Brno, Czech Republic) with high-resolution imaging. Element mapping was carried out in the scanned area by energy dispersive spectroscopy (EDS, Oxford Instruments, Concord, MA, USA). The dispersion of bentonite particles in the PE wax matrix was investigated by a back scattered electron (BSE) signal and a secondary electron (SE) signal with an accelerating voltage of 10 kV. A magnification of 1000× was used.

### 3.6. Fourier Transform Infrared (FT-IR) Spectroscopy

Changes in the chemical structure of polyethylene wax caused by incorporating bentonite were evaluated on the basis of Fourier transform infrared spectroscopy. Fourier transform infrared (FT-IR) spectroscopy analysis was carried out using a Jasco FT/IR 4600 spectrometer (Jasco Europe S.R.L., Cremella, Italy). The spectra were collected over the wavelength range of 4000–400 cm^–1^ at room temperature, with a resolution of 4 cm^–1^, after averaging 40 scans. Spectroscopic data were treated using the dedicated software Spectra Manager (ver. 2, Jasco, Easton, MD, US).

## 4. Results and Discussion

### 4.1. Effect of the Bentonite on Pure PE-Wax Rheological Behavior

Figure 7 presents the dynamic viscosity values measured at a constant shear rate as a function of time of PE-wax/BNT blends. The figure indicates a slight decrease in viscosity at low content bentonite up to 0.4 wt.%. The tendency was significantly changed for samples containing 1 and 5 wt.% of the filler. For higher than 1 wt.% of BNT, the apparent viscosity gradually increased with increasing filler content in the PE-wax matrix. However, the addition of 5% by weight bentonite has increased the viscosity compared to pure PE-wax. This results in an increase in the efficiency of perfect bonding between PE-wax and bentonite, which consequently gives rise to higher hardness. Furthermore, Chakravorty at al. [13] observed effects for cross-linked polystyrene-filled hydrocarbon wax, an emulsified pattern wax, and an acid-filled pattern used in the investment casting industry, where the viscosity in all cases decreased with increasing shear rate and, for a given temperature viscosity, all waxes broadly varied within an order of magnitude [13].

Figure 8 shows the flow (a) and viscosity curves (b) of polyethylene waxes filled with different bentonite contents. The flow curves (Figure 8a) show the modification of the flow character of PE-wax induced by the addition of the BNT. The pure PE-wax as well as compositions containing 0.1 and 0.4 wt.% shows almost no change of the flow curves shape. However, the addition of more than 0.4 wt.% of BNT leads to the creation of a yield point of the molten waxes, the more distinct the higher amount of the filler is. It is assumed that the dispersions with yield points of above 10 Pa have sufficient structural strength at rest to show physical stability against sedimentation of the filler, in case of its limited density. Therefore, it can be stated that the change of the rheological character of the PE wax may suppress potential negative effects of the sedimentation of particle-shaped filler in case of its higher amount of addition. Unmodified PE-wax, similarly to compositions containing 0.1 and 0.4 wt.% of BNT, shows no influence of shear rate on viscosity. In the case of compositions containing 0.8 wt.% BNT and more, the phenomenon of shear-thinning behavior and a significant increase in viscosity of filled wax at a low shear rate are observed, which was the expected effect, according to previously published results [49]. The significantly higher viscosity values at low shear rates are likely the result of the occurrence of agglomerated filler structures. Along with the increase in shear rate during the test, filler directs toward the flow, which can be observed by limiting the relationship between viscosity and shear rate. Discrepancies in measured viscosity values at constant (Figure 7) and logarithmically increasing (Figure 8) shear rate conditions result from the application of the initial shear process for the first test type, which caused the plate filler to be oriented and partially broken down, which, thereby, hindered filler structures for compositions containing 1 and 5 wt.% of the BNT.

The results of density tests (Table 5) show that the addition of bentonite with a content of up to 5 wt.% does not significantly change the density of the tested wax blends used in the lost-wax method.

### 4.2. The Effect of Bentonite on Linear Shrinkage of PE-Wax

The results of linear shrinkage (SL) of pure wax and blends that were made from polyethylene wax with different bentonite content are shown in Figure 9.

Generally, it can be seen that the linear shrinkage for all wax–bentonite blends decreases when compared to unmodified wax. The results showed that the smallest shrinkage (approx. 0.41 ± 0.10%) was recorded for a wax mixture with 0.8 wt.% bentonite. Moreover, a significant change of the SL for PE-wax was shown with a 0.4 wt.% addition of bentonite. In this case, the linear shrinkage reduced from 2.65 ± 0.05% for pure wax to a value of 0.55 ± 0.15%, which means that shrinkage of this wax/0.4 BNT blend is lower by approximately 90% when compared to pure wax. The increase in shrinkage for mixtures with bentonite content without chemical modification to 1 wt.% is promoted by an increase in the degree of crystallinity. Thin PE-wax samples can have a very strong macromolecular orientation and systolic anisotropy. Furthermore, it was found that a 5 wt.% concentration of the filler in wax made no improvement to the linear shrinkage, as compared to samples where the filler content was 1 wt.%. This may show that a certain limit of concentration on the modified bentonite in the PE-wax has been reached.

Previous research [50] showed that the linear shrinkage for pure waxes, e.g., Aqua, Red, and Blue waxes, produced by the same manufacturer, increased with the increasing number of melting cycles. In this paper, the influence of multiple re-melting of wax blends on thermal properties and linear shrinkage was analyzed in order to verify the possibility of their reuse in the production of precise models. Lastly, it also showed that a constant increase in the linear shrinkage with a rising number of melting cycles is a disadvantage due to inaccuracy in the mapping of models constructed using dies at ambient temperature [50]. A similar phenomenon was confirmed by Fisher [51] who proved that plastics with higher density, and, particularly polyethylene, show more significant processing shrinkage and materials with lower density have lower ability to create crystalline structures due to the strong branching of the chain. Usually, the shrinkage of polyethylene wax, as well as semi-crystalline polymers, decreases with the addition of particle-shaped fillers. Bentonite introduced in the form of powder significantly affects the change of casting conditions by the smelted model method. In this case, it showed a positive effect because it caused a decrease in linear shrinkage and dimensional stabilization of PE-wax/BNT blends. Therefore, it affects the quality of the resulting casting without causing clinging errors and warping [1,2,3,4,5,52,53,54].

### 4.3. The Effect of Bentonite Content on the Hardness of Pure PE-Wax

Figure 10 shows the dependence of PE-wax blends’ hardness on the BNT content. Generally, results indicate that filled compositions exhibit decreased hardness when compared to pure wax. The lowest hardness of 14.5 ± 0.5 °ShD was obtained for a wax blend with 0.4 wt.% BNT content, which means that hardness of this blend is lower by approximately 30% when compared to unmodified PE-wax. These results indicate that the PE-wax becomes more flexible under the influence of the BNT in the range of 0.1 to 0.8 wt.% added in this study. However, for a blend containing 5 wt.% of BNT in the PE-wax matrix, a significant increase in hardness to about 20 °ShD was observed, which corresponds to the hardness of pure wax. In the case of this wax/BNT blend, a change in color from blue to dark grey was observed. This is likely due to the magnesium and calcium reaction of the chemical binding with atoms of the hydrocarbon chain of wax [27]. The hardness results obtained correlate with linear shrinkage changes for the wax/BNT blend.

### 4.4. The Effect of Bentonite Content on the PE-Wax Thermal Properties

The effectiveness of wax modification realized by incorporating bentonite was evaluated using DSC. The selected thermal parameters obtained from DSC are summarized in Table 6.

Figure 11 presents the DSC thermograms for pure PE-wax and its blends with different bentonite content obtained during the second heating (Figure 11a) and the second cooling cycle (Figure 11b). On DSC melting curves, there is a only one distinct endothermic peak with a maximum at 60 °C, which corresponds to the melting point of PE-wax [35,55]. The lack of additional peaks as well as inflections of the DSC curve during the melting and cooling cycle caused by the addition of BNT may be attributed to the lack of influence of the filler on a form of the PE-wax crystalline domain structure. This effect should also be attributed to the good dispersion of the particle-shaped filler in the polyethylene wax matrix. As mentioned, the addition of bentonite in the range from 0.1 to 0.4 wt.% did not provide significant changes of PE-wax/BNT blends’ viscosity. Therefore, it was possible to achieve proper dispersion of bentonite in PE-wax. A similar phenomenon for the pure polyoxymethylene and its composites was observed, which indicated that, as the result of the modification, compatible blends of polyoxymethylene (POM) with polysilsesquioxane (POSS) particles were achieved [56,57,58,59].

On the basis of DSC results, it was found that the melting onset temperature (*T_m_*
_onset_) from the second heating cycle for wax blends with 1 wt.% bentonite was higher by approximately 7 °C compared to unmodified PE-wax. Similarly, the enthalpy of melting from the second cycle is significantly higher for the PE-wax/1 BNT mixture, which indicates the modifying interaction of bentonite on the polyethylene wax matrix. The addition of low bentonite content up to 1% by weight slightly increases the crystallization temperature of the blends. Therefore, it can be stated that a nucleating efficiency of BNT on PE-wax is limited. The opposite effects were observed by Othman et al. [33] for polypropylene-filled bentonite composites. Compared with pure PP, the bentonite filled-PP composites had lower crystallization and melting temperature, lower crystallization enthalpy, and narrower and sharper exothermic peak course during cooling [33].

The improved crystallinity can be observed for PE-wax modified by bentonite in the range of 0.1–1 wt.%. A significant change of the crystallinity for PE-wax with a 0.4 wt.% addition of bentonite was shown. In this case, the crystallinity increased to value 46.7 ± 0.7% from 41.5 ± 0.5% for pure wax, which means that the *Xc* of wax/0.4 BNT blend is higher by approximately 11% when compared to pure wax. This means that the bentonite indicates that, despite the incorporation of BNT causing the enhancement of the crystallization kinetics of PE-wax, the dispersed rigid structures of the filler may act as active nucleation centers and increase the amount of the nuclei in PE-wax melt during solidification [38,47]. Although the increase in the degree of crystallization of polymers is usually accompanied by higher shrinkage. In the considered case, the effect of the presence of rigid inorganic filler structures was dominant over the crystallinity increase and acted to suppress the linear shrinkage of PE-wax/BNT blends. In general, an increase in melt temperature is usually accompanied by decreased shrinkage of final products likely due to lower viscosity of the melt that promotes more uniform pressure distribution of the filler in the polymeric matrix as well as the material throughout the mold. A similar finding was also observed by Razavand et al. [22]. They found that both melt temperature and holding time have great influences on the final dimensions of injected wax patterns, such as in gas turbine blades [22].

### 4.5. The Effect of Bentonite on the Morphology PE Wax/BNT Blends

SEM microscopy was used to analyze the surface morphology and dispersion of particles of BNT in the PE-wax. Figure 12 shows the morphology SEM images of pure wax and PE-wax/BNT blends made using secondary electron (SE) signals (left side) and an image prepared with a back-scattered electron (BSE) detector (right side). BSE comes from deeper regions of the sample while SE originates from surface regions. Therefore, BSE and SE carry different types of information about the PE-wax/BNT blend. BSE images show high sensitivity to differences in atomic number. The higher the atomic number, the brighter the material appears in the image while SE imaging can provide more detailed surface information [60]. SEM micrographs of PE-wax blends showed significant differences in the morphology of the compositions between pure PE-wax (Figure 12a,b) and were filled with various amounts of the filler compositions. As it follows from Figure 12c,d, at a low content of 0.8 wt.% of the bentonite, a relatively homogenous distribution of BNT in the wax matrix may be observed, which indicates the miscibility of BNT within the PE-wax matrix. Good miscibility of wax/0.1–0.4 BNT blends was a result of reduced melt viscosity. In the case of a higher bentonite amount (above 1 wt.%), the filler demonstrated a tendency to create agglomerates, as shown in Figure 12h–l. For the PE-wax blend with 5 wt.% in the Scanning Electron Microscope with a Back-Scattered Electron (SEM-BSE) image, the agglomerates with dimensions from 10 to 40 µm (marked with red circles) in the matrix could be seen in Figure 12l. This proves the heterogeneous dispersion of a large amount (above 1% by weight) of bentonite in the polyethylene wax matrix due to an excessive melt viscosity. It should be mentioned with regard to bentonite-modified wax blends, no sedimentation process of BNT particles in PE-wax was observed, which is often the case for blends with a large difference in medium density and filler [3,4,53].

### 4.6. FT-IR Characterization of Pure PE-Wax Filled With Bentonite

FT-IR analysis was conducted to obtain identified and detailed pure wax structural information and examine the bond interaction within the bentonite and PE wax blends. Figure 13 presents the Fourier-transform infrared spectroscopy (FT-IR) spectra of pure PE-wax and wax/1BNT blend (red curve). As shown, pure PE wax shows its characteristic peaks at 2915 cm^–1^ and 2848 cm^–1^ associated with the symmetrical and asymmetrical aliphatic –CH stretching vibration of carbon-hydrogen bonds [33,53,61,62,63], which confirms saturated chains. The weak absorbance at 1720 cm^–1^ relates to the carbonyl stretching vibration of the ester group (–C=O stretch). The absorption at 1463 cm^–1^ corresponds to the stretching vibration attributed to the CH_2_ bond (–CH_2_ scissoring) and 1377 cm^–1^ (–CH_3_ umbrella mode), and the band at 720 cm^–1^ (–C–C rocking) refers to the stretching vibration of the carbon-carbon bond. Similar spectra were reported by Thainá Araújo de Oliveira et al. when evaluating the influence of Carnauba wax on films of poly (butylene adipate co-terephthalate) and sugarcane residue for application in soil covers [61].

Figure 13 (red curve) shows the FT-IR spectrum of the PE-wax modified with bentonite. As observed, three new peaks at 3440, 1638, and 1043 cm^–1^ appeared, which could be attributed to bentonite adsorption on the PE-wax interface. A weak band at 3440 cm^–1^ is due to the hydroxyl group of the bentonite structure [63]. The strong peak near 1638 cm^–1^ can be attributed to the interaction between the organic molecule and the bentonite. The signal at 1043 cm^–1^ characterizes the silicate groups of the bentonite [63]. The results suggest that the interaction between the polyethylene groups and the hydroxyl groups of the bentonite are responsible of properties’ improvement. In addition, the bands at 1100 and 1040 cm^−1^ are identified with the symmetric and asymmetric stretching of the C–O bond of an ester [64,65]. A similar effect was observed by Lozechnikowa [62] in Carnauba wax dispersion on wood. The signal at 1015 cm^−1^ characterizes the silicate groups of the bentonite. Lastly, the band at 860 cm^−1^ is related to silicates [63]. The FT-IR spectrum of bentonite shows the bands at 653 cm^−1^ for Al–OH, 784 cm^−1^ due to (Al, Mg)–OH vibration modes, and 515 and 458 cm^−1^ for Si–O bending [33,53]. The FT-IR spectra showed that the interaction of bentonite filler with PE-wax as possible.

### 4.7. Assessment Casting Produced by the Lost-Wax Method

As a result of the lost-wax casting process, very good mapping of the cast shape was obtained. The resulting castings are shown in Figure 14. Final products were characterized by excellent quality, especially the low roughness of the properly restored surface. After initial macroscopic examination, no shape defects, raw casting surface defects, or discontinuities were found. Furthermore, no defects, such as a systolic cavity, rash, or blisters belonging to internal defects, were registered. [3,55,56,66,67,68,69]. Further analysis of the quality of silumin castings made using the smelted model method and the proposed mixture for the PE-wax/bentonite model is not included in the scope of this work, but will be the focus of the authors’ research in subsequent publications. The aim of this paper was to present the possibilities of applying the new material, i.e., polyethylene wax modified by bentonite to produce the models in the foundry and determine if the created casting (Figure 14) would be acceptable and properly formed without traces of bentonite particles on the casting surface. The presented cast in Figure 14 is a test cast, which, in the future, will be subjected to an assessment of its structure, chemical composition, and quality using the FMA (failure mode and effects analysis) method and Ishikawa diagram.

## 5. Conclusions

A new material, based on polyethylene wax modified with bentonite, was produced, and the possibilities for applying the new material to produce models in a foundry were proposed. The influences of various bentonite content levels on the viscosity, linear shrinkage, crystallinity, morphology, and hardness of PE-wax used in the precision casting process was investigated.

The viscosity results indicate a slight decrease in viscosity at low content of bentonite up to 0.8 wt.%. For higher than 1 wt.% of BNT, the apparent viscosity gradually increased with increasing filler content in the PE-wax matrix. It was found that the addition of 0.4 wt.% bentonite to PE-wax caused the most significant reduction of the linear shrinkage by approximately 90%, in comparison to unfilled wax, stabilizing the shrinkage value to about 0.5%. Generally, hardness results indicate that bentonite-filled wax blends exhibit decreased hardness when compared to pure wax. These results show that the PE-wax becomes more flexible under the influence of BNT in the range of 0.1 to 1 wt.% added in this study. The hardness of wax with 0.4 wt.% BNT content is 30% lower than that of unmodified PE-wax.

DSC studies indicate good miscibility of wax with bentonite, as shown by the presence of one melting point. This is also confirmed by microscopic images (SEM). For a mixture of PE-wax with 0.4 wt.% BNT, a significant increase in the melting temperature and enthalpy was observed when compared to the pure wax. The crystallinity of the wax/0.4BNT blend increased by approximately 11% when compared to the unmodified wax. The effect obtained is the result of an interaction of bentonite as a crystallization modifier for unmodified wax. Based on SEM microscopic observations of the PE-wax/BNT mixture, no signs of brittle fracture were observed, and only a plastic deformation area was found.

Moreover, the following conclusions can be drawn from the investigations conducted on PE-wax/BNT.

Bentonite is an effective modifier of linear shrinkage, hardness, and crystallinity in addition to polyethylene wax, when used in lost-wax casting, as distinguished by the lack of sedimentation of the filler from polyethylene wax.Modification with bentonite leads to a reduction of linear shrinkage by approximately 90% and hardness by 30% when compared to pure PE-wax, and, thus, improves the dimensional accuracy of wax models and the casting process.The new material, based on polyethylene wax modified with bentonite ranging from 0.1 to 0.4 wt.%, results in reduced shrinkage and only a slight increase in viscosity, which, thus, suits it perfectly to lost-wax casting.Aluminum alloy castings with excellent surface quality were obtained, as confirmed by macroscopic research based on observing the casting surface with the naked eye or using a magnifying glass. These castings were characterized by a smooth surface. There were no defects in the shape or raw surface, or breaks in continuity. Furthermore, no internal defects such as systolic cavities, scabs, or blisters were recorded.

## Figures and Tables

**Figure 1 materials-13-02255-f001:**
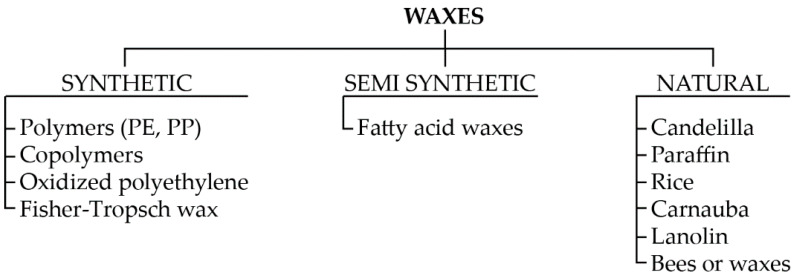
Division scheme of waxes.

**Figure 2 materials-13-02255-f002:**
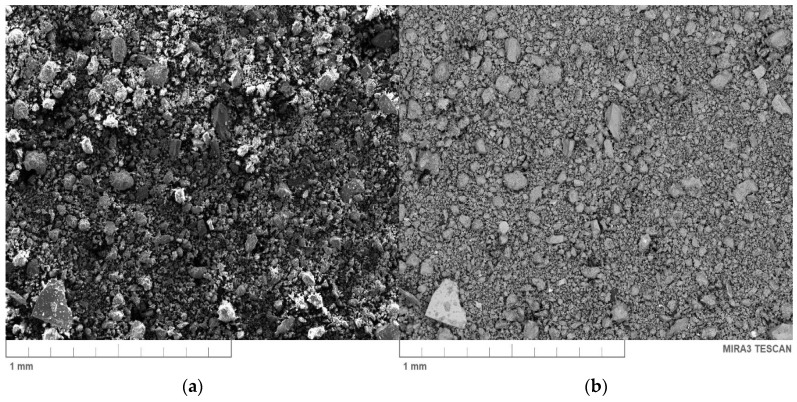
Scanning electron microscopy (SEM) images of a bentonite: (**a**) secondary electron (SE) (**b**) and back-scattered electron (BSE).

**Figure 3 materials-13-02255-f003:**
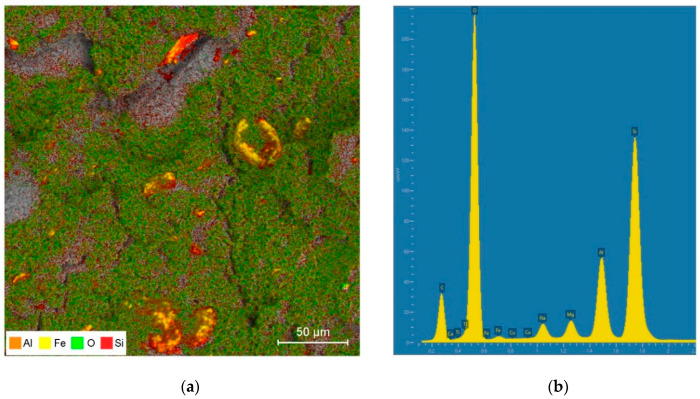
(**a**) Energy dispersive spectroscopy (EDS) mapping, and (**b**) the distribution of chemical composition of bentonite.

**Figure 4 materials-13-02255-f004:**
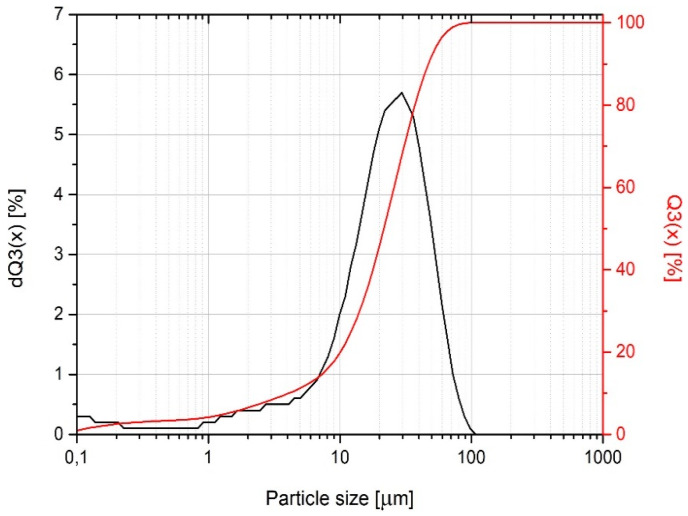
The particle size distribution of bentonite used as a filler.

**Figure 5 materials-13-02255-f005:**
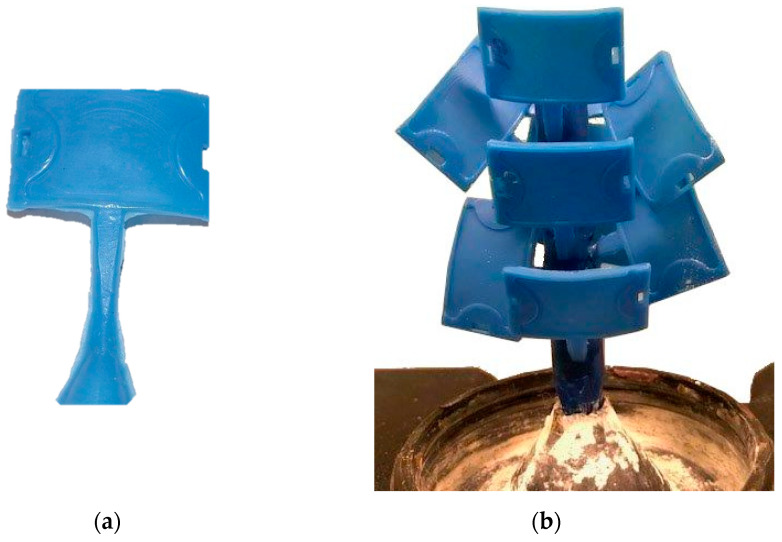
PE-wax/0.4 bentonite (BNT) system: (**a**) wax pattern, (**b**) pattern assembled on a “tree.”

**Figure 6 materials-13-02255-f006:**
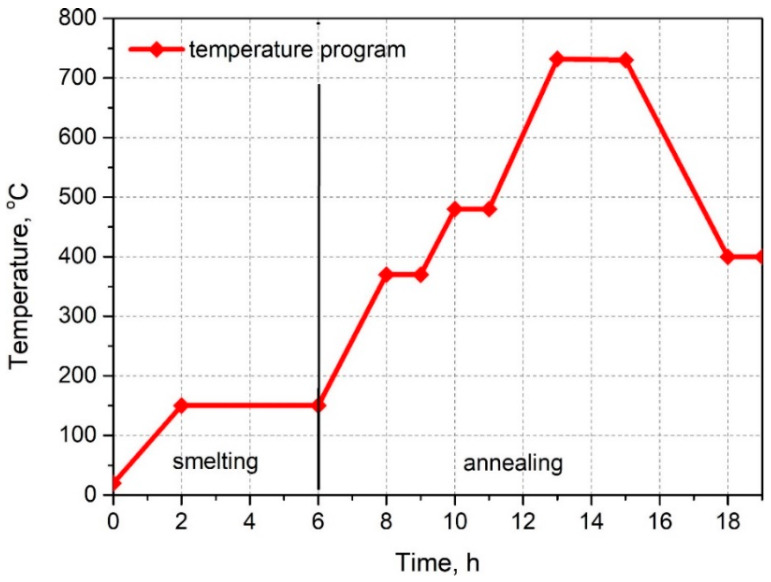
The smelting process and heat treatment (annealing) of the gypsum mold.

**Figure 7 materials-13-02255-f007:**
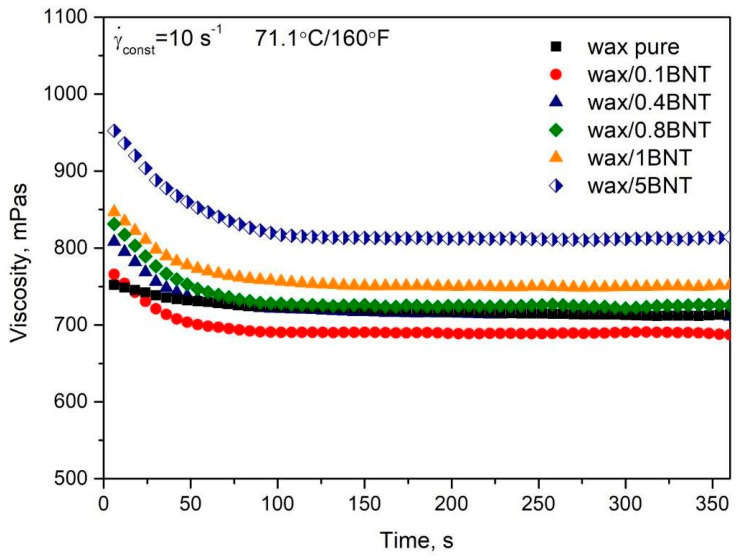
The viscosity measurements of pure polyethylene wax filled with various amounts of bentonite measured at constant shearing conditions (10 s^–1^).

**Figure 8 materials-13-02255-f008:**
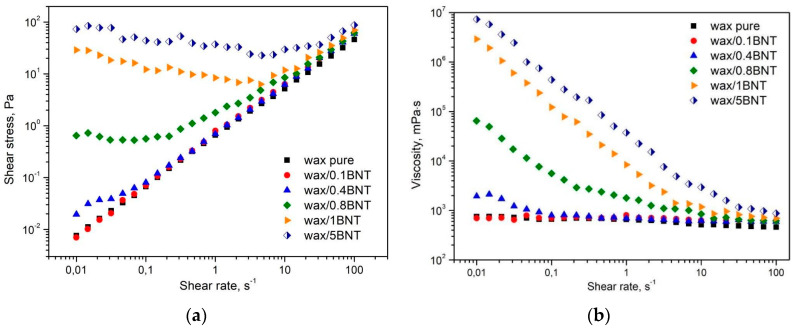
The flow (**a**) and viscosity curves (**b**) of pure PE-wax filled with various amounts of bentonite.

**Figure 9 materials-13-02255-f009:**
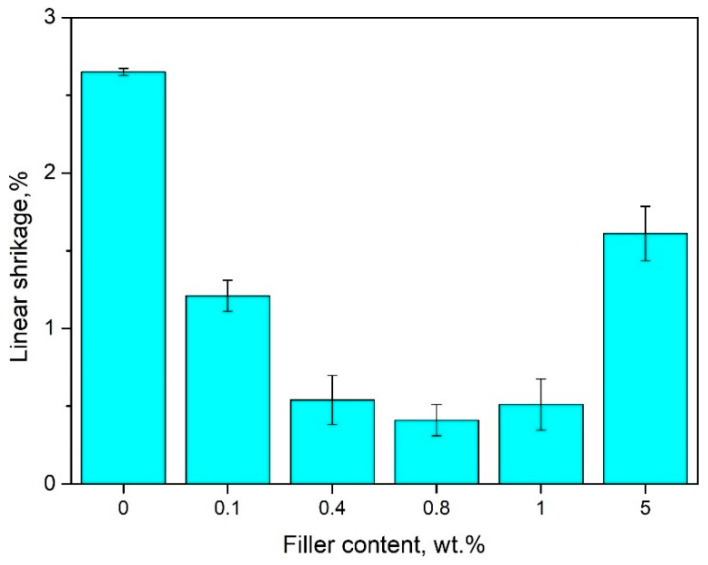
Linear shrinkage of pure PE-wax filled with various amounts of bentonite.

**Figure 10 materials-13-02255-f010:**
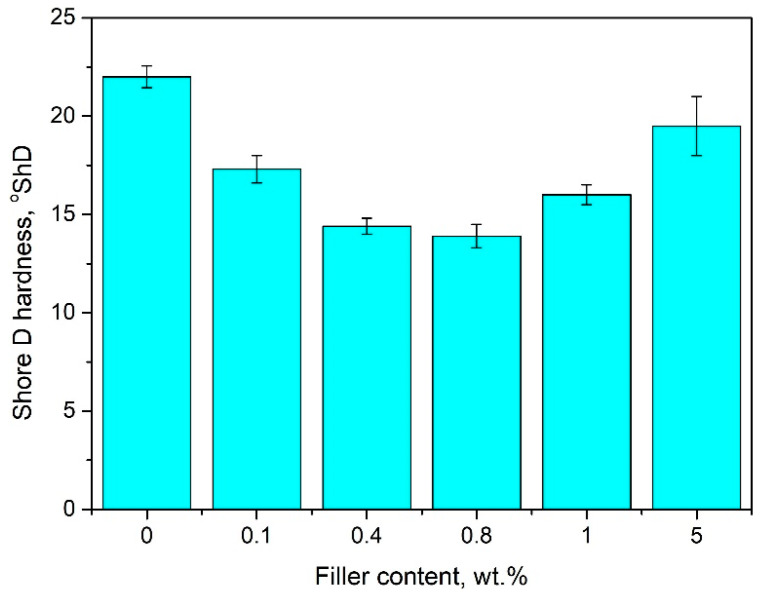
Shore D hardness of pure PE-wax filled with various amounts of bentonite.

**Figure 11 materials-13-02255-f011:**
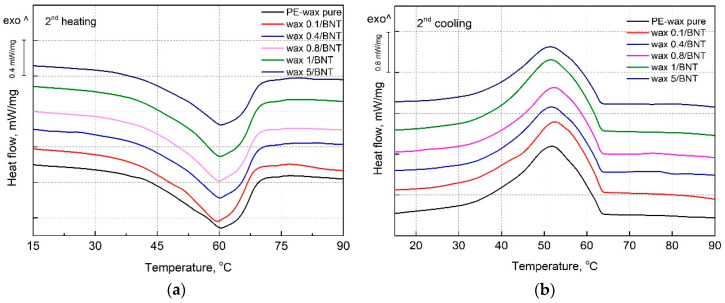
Differential scanning calorimetry (DSC) melting (**a**) and crystallization (**b**) curves of pure PE-wax filled with various amounts of the bentonite.

**Figure 12 materials-13-02255-f012:**
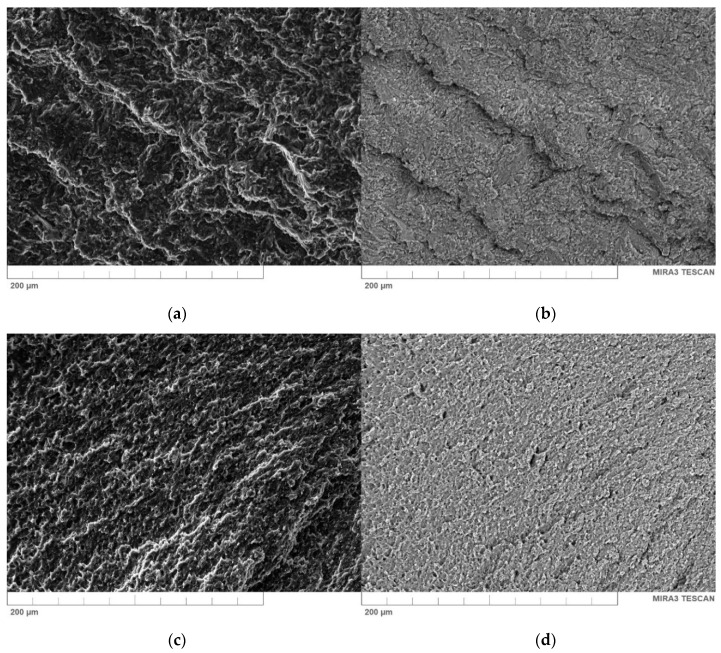
Scanning electron microscopy (SEM) images of fractured surface: (**a**,**b**) pure PE-wax, (**c**,**d**) wax/0.1BNT, (**e**,**f**) wax/0.4BNT, (**g**,**h**)wax/0.8BNT, (**i**,**j**) wax/1BNT, (**k**,**l**) wax/5BNT (magnification 1000×), SE (left) and BSE (right).

**Figure 13 materials-13-02255-f013:**
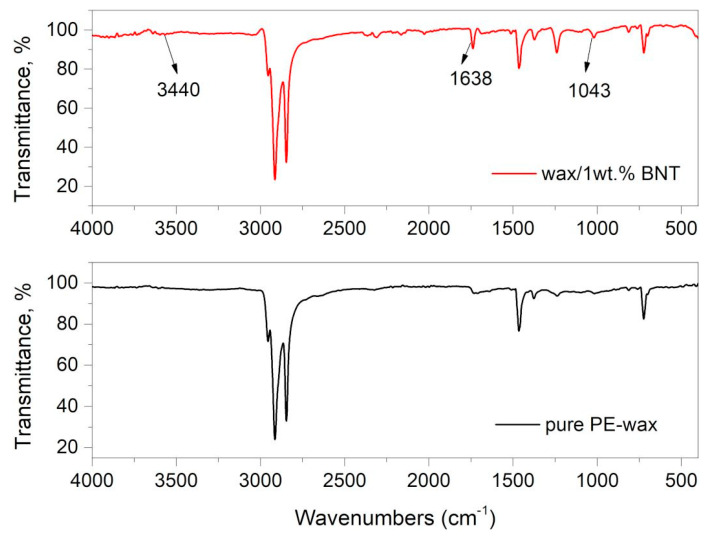
FT-IR spectra of PE-wax with 1 wt.% BNT blend (red curve) and pure PE-wax.

**Figure 14 materials-13-02255-f014:**
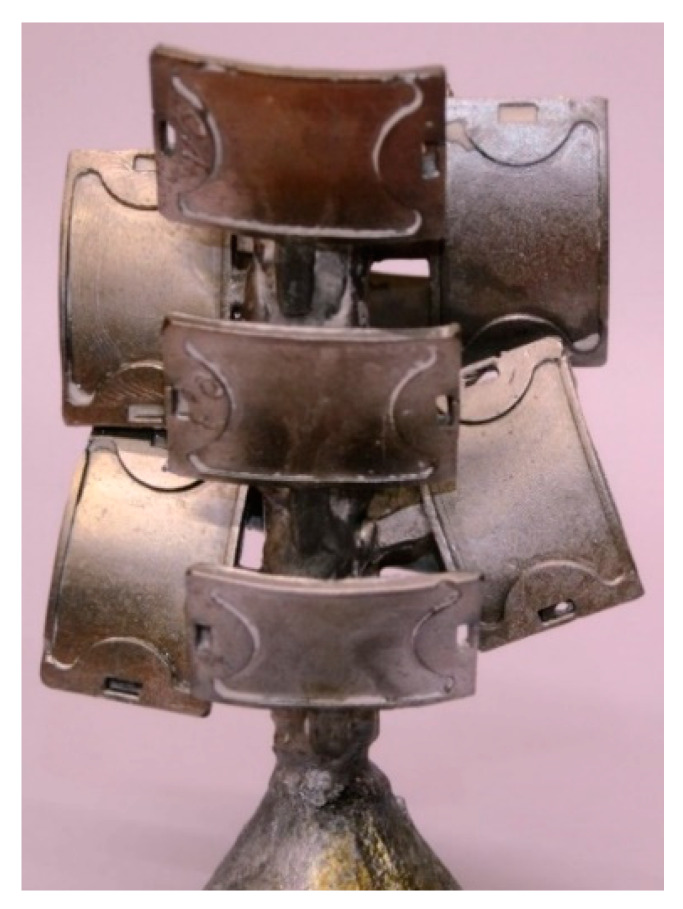
An example of the application of PE-wax modified by bentonite for the production of an aluminum alloy casting in the lost-wax casting process.

**Table 1 materials-13-02255-t001:** Selected physical properties of flexible polyethylene (PE) wax [44].

Properties	Viscosity (160 °F)/71.1 °C (mPa·s)	Injection Temperature (°F/°C)	Geometrical Form	Shore D Hardness (°ShD)
Flexible Blue Wax	615	170/76.7	flakes	30

**Table 2 materials-13-02255-t002:** Chemical composition of bentonite in wt.%.

Element	O	Si	Al	Fe	Ca	Mg	Na	K
wt.%	49.2	30.4	10.4	2.9	2.1	2.0	1.8	0.7

**Table 3 materials-13-02255-t003:** Recipes used for bentonite-filled polyethylene wax (PE-wax).

Sample	Bentonite (wt.%)	Bentonite (vol.%)
wax pure	0	0
wax/0.1BNT	0.1	0.13
wax/0.4BNT	0.4	0.5
wax/0.8BNT	0.8	1
wax/1BNT	1	1.25
wax/5BNT	5	6.25

**Table 4 materials-13-02255-t004:** The injection process parameters of pure wax and wax/BNT blends.

Parameters	Pure Wax	Wax/BNT Blends
Wax temperature, °C	72	73
Injection temperature, °C	73	74
Injection pressure, kPa	30	31
Holding time, s	30	32

**Table 5 materials-13-02255-t005:** Results of density measurements of PE-wax and its PE-wax/BNT blends.

Sample	Density, g/cm^3^
wax/0.1BNT	1.026 ± 0.001
wax/0.4BNT	1.047 ± 0.002
wax/0.8BNT	1.049 ± 0.003
wax/1BNT	1.059 ± 0.002
wax/5BNT	1.070 ± 0.005

**Table 6 materials-13-02255-t006:** DSC data of pure PE-wax and its blends with various content of the bentonite.

Samples	Onset Melting Temperature *T_m_* _onset_ [°C]	Melting Temperature *T_m_*_2_ [°C]	Heat of Fusion Δ*T_m_*_2_ [J/g]	Crystallization Temperature *T_cr_*_2_ [°C]	Degree of Crystallinity *X_c_* [%]
wax pure	42.0	60.0	−123.2	51.0	41.5 ± 0.5
wax/0.1BNT	45.6	60.3	−135.4	52.2	45.6 ± 0.4
wax/0.4BNT	47.6	60.4	−138.0	52.4	46.7 ± 0.7
wax/0.8BNT	47.3	60.0	−137.0	52.3	46.5 ± 0.8
wax/1BNT	48.6	60.2	−139.4	52.0	47.4 ± 1.0
wax/5BNT	44.6	60.4	−127.0	52.0	43.6 ± 1.2

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
