# Peer review of "Polyethylene Wax Modified by Organoclay Bentonite Used in the Lost-Wax Casting Process: Processing−Structure−Property Relationships"

_materials, 2020, doi:10.3390/ma13102255_

Round 1

Reviewer 1 Report

The aim of this research is to develop new was formulation to reduces the linear shrinkage and thermal resistances, and hopefully improved the quality of aluminum cast products.

Experimental procedures seem sufficient to accomplish the paper objectives, and discussion was found relevant.  However, some issues need to be addressed and considered 

  1. Impro"ve English language. Line 52 "This wax has a melting point about 90ºC and a TEC of..."
  2. The introduction needs some fixing. Wax types and properties must be group together
  3. Line 75,76: You need to explain in more details the detrimental phenomenons and add some references
  4. Wax properties are link to the additives and wax formulation. Does the chemical purity has a relevant effect? Please discusses this briefly.
  5. The introduction must be resume. It is too long and explains aspects far from the paper objectives.
  6. Table 1. Use SI unit system for wax properties. Viscosity has no units.
  7. Table 2. Replaces "An elements" by "Element"
  8.  BNT means bentonite?. Please clarify it.
  9. If 1-5 wt% of BNT was added, how much in volume it represents? Pleases include this information.
  10. If the wax mixtures were injected, how the BNT concentration was retained? Did you confirm there is no formation of BNT concentration gradient?
  11. Characterization Methods: 3.1 and 3.2 sections need some attention. Please remember, you need to be explained your methods sufficiently, so others can replicate yours. Pleases give more details especially on the measurement of viscosity.
  12. Line 292. "Degree of crystallinity Xc of the samples was determined from the values of enthalpy..." Could you give a reference to this procedure? I do not think the DSC technique suitable to evaluated crystallinity by DSC. If so, could you replace the reference 41 for some other?
  13. Figure 6. Usually, viscosity values are represented with log graphs. Since you have a rheometer, please add more experimental data on the viscosity behavior of the wax.
  14. Section 4.7 Great effort to make an aluminum cast part. However, quality evaluation was obtained purely by observation, no comparison was made with another metal part, for example using a wax with no filler contend. I do not think this part is relevant, pleases consider its elimination.  

Reviewer 2 Report

See attachment

Round 2

Reviewer 1 Report

There are some details still need to be addressed. I numbered according to your cover letter response.

1. Line 59. .... as systematized in Figure (missing the number 1)

2-5. The introduction is still very long. It is clear your knowledge is extensive, and you master the literature, but perhaps you need to center on the main topic relating to your research.

Wax reuse is an important topic, keep it.

The additives dissertation must be summarized.

6-8. Corrected 

9. Pleases add the wax and bentonite densities values in section 2.1.

10. Agree. BNT particles are under 200 microns and wax viscosity is somehow high, more than 600 mPa·s. 

11. Additional information regarding viscosity measurements seems sufficient.

12. The use of DSC technique for the determination of the degree of crystallinity is now well explained and supported. 

13. Since wax is a non-newtonian fluid, it is more suitable to present the graphs as the shear stress (Y) versus the shear rate (X). Please modify Figures 7 and 8 accordingly. 

From Figure 8, should be more clear to show the effect on BNT additions on the rheological behavior. 

14. Section 4.7 add more value to the manuscripts.

Figure 7 will give you information at constant shear rate, 

again Figure 7 and Figure 8 as 

Author Response

Response to Reviewer 1

The authors thank the reviewer for their thorough evaluation and constructive comments on the manuscript. All responses to Review 1 have been included below and in the text by the authors.

English language and style were improved by MDPI English Editing.

Line 59. .... as systematized in Figure (missing the number 1)

This has been added. (…in Figure 1).

2-5. The introduction is still very long. It is clear your knowledge is extensive, and you master the literature, but perhaps you need to center on the main topic relating to your research.

Wax reuse is an important topic, keep it.

The additives dissertation must be summarized.

The authors corrected a too-long introduction to the article. We focused on the main topic relating to wax modified by bentonite (changes see in a manuscript).

6-8. Corrected 

  1. Pleases add the wax and bentonite densities values in section 2.1.

This has been added in section 2.1.

  1. Agree. BNT particles are under 200 microns and wax viscosity is somehow high, more than 600 mPa·s. 
  2. Additional information regarding viscosity measurements seems sufficient.
  3. The use of DSC technique for the determination of the degree of crystallinity is now well explained and supported. 
  4. Since wax is a non-newtonian fluid, it is more suitable to present the graphs as the shear stress (Y) versus the shear rate (X). Please modify Figures 7 and 8 accordingly. 

From Figure 8, should be more clear to show the effect on BNT additions on the rheological behavior. 

- Figure 8 presents dynamic viscosity curves achieved during the experiment conducted under the logarithmically step increasing shear rate. While, in general wax is a non-Newtonian fluid, which shows shear thinning behavior at higher shear rates, in considered range of low shear rated the Newtonian flow region was observed. The distinct impact of BNT filler on rheological behavior may be observed in Fig. 8, and it was described in the revised version of the manuscript. The unmodified PE wax reveals almost Newtonian behavior in measured shear rate range, as it was also observed for the wax/0.1BNT sample. Incorporation of the inorganic particle filler with amount 0.4 wt% change slightly the rheological behavior of the composition, while further higher BNT content caused the significant increase of the dynamic viscosity at lower shear rate values, the more distinct, the higher concentration of the BNT. As it was described in the revised version of the manuscript, this is a typical effect of the creation of agglomerated structures of the filler. The rigid particle domains dispersed in the molten PE wax became in a mutual contact, which causes the observed at viscosity curves the significant increase of the viscosity. With increasing shear rate hindered inorganic particles agglomerates were subjected to movement, and filler particles were dispersed in the polymeric matrix in the direction of the forced flow. As an effect, for the considered with higher BNT content, the similarity and coverage with the flow curve of unmodified PE wax were achieved only at higher shear rates.

- The additional flow curve, according to Reviewers comment, was included in Figure 8. Figure 7 shows the experiment conducted in order to achieve the viscosity value comparable to those presented in the technical datasheet, and this is why the column plots were presented instead of viscosity or flow curves. From the point of view of the application of developed materials, the presentation of flow curves, in case of a constant shear experiment, will not provide additional valuable information about the rheological behavior of the compositions. Especially that the dynamic viscosity is the quotient of shear stress and shear rate. However, we change the presentation of the viscosity values determined at constant shear.

- More information about viscoelastic behavior may be achieved by the realization of frequency and strain sweep experiments; at this time, unfortunately, it cannot be conducted in our laboratory using measuring set-up.

Reviewer 2 Report

See attachment

Author Response

We cannot respond to the Reviewer's comments because there is no attachment. Please add an attachment.
